# Underlying Molecular Mechanism and Construction of a miRNA-Gene Network in Idiopathic Pulmonary Fibrosis by Bioinformatics

**DOI:** 10.3390/ijms241713305

**Published:** 2023-08-27

**Authors:** Shuping Zheng, Yan Zhang, Yangfan Hou, Hongxin Li, Jin He, Hongyan Zhao, Xiuzhen Sun, Yun Liu

**Affiliations:** Department of Respiratory and Critical Care Medicine, The Second Affiliated Hospital of Xi’an Jiaotong University, Xi’an 710004, China; zsp3117331013@stu.xjtu.edu.cn (S.Z.); zy578180410@stu.xjtu.edu.cn (Y.Z.); hyf19930226@stu.xjtu.edu.cn (Y.H.); nblhx123@stu.xjtu.edu.cn (H.L.); jin.he@stu.xjtu.edu.cn (J.H.); 3121315361@stu.xjtu.edu.cn (H.Z.); doc-ly@sohu.com (X.S.)

**Keywords:** idiopathic pulmonary fibrosis (IPF), bioinformatics, differentially expressed genes (DEGs), miRNAs (microRNAs)

## Abstract

Idiopathic pulmonary fibrosis (IPF) is a chronic, progressive lung disease, but its pathogenesis is still unclear. Bioinformatics methods were used to explore the differentially expressed genes (DEGs) and to elucidate the pathogenesis of IPF at the genetic level. The microarray datasets GSE110147 and GSE53845 were downloaded from the Gene Expression Omnibus (GEO) database and analyzed using GEO2R to obtain the DEGs. The DEGs were further analyzed for Gene Ontology (GO) and Kyoto Encyclopedia of Genomes (KEGG) pathway enrichment using the DAVID database. Then, using the STRING database and Cytoscape, a protein–protein interaction (PPI) network was created and the hub genes were selected. In addition, lung tissue from a mouse model was validated. Lastly, the network between the target microRNAs (miRNAs) and the hub genes was constructed with NetworkAnalyst. A summary of 240 genes were identified as DEGs, and functional analysis highlighted their role in cell adhesion molecules and ECM–receptor interactions in IPF. In addition, eight hub genes were selected. Four of these hub genes (*VCAM1*, *CDH2*, *SPP1*, and *POSTN*) were screened for animal validation. The IHC and RT-qPCR of lung tissue from a mouse model confirmed the results above. Then, miR-181b-5p, miR-4262, and miR-155-5p were predicted as possible key miRNAs. Eight hub genes may play a key role in the development of IPF. Four of the hub genes were validated in animal experiments. MiR-181b-5p, miR-4262, and miR-155-5p may be involved in the pathophysiological processes of IPF by interacting with hub genes.

## 1. Introduction

Idiopathic pulmonary fibrosis (IPF) is a chronic and progressive lung disease [1], which is characterized by the development of diffuse, progressively remodeled lung parenchyma, the deposition of extracellular matrix, and the formation of irreversible scarring [2], eventually resulting in breathing difficulties and even death [1]. It is a fibrotic chronic interstitial pulmonary disease of unknown origin. The disease is prevalent in middle-aged and older people [3], its incidence has increased dramatically in recent years [4], and its prognosis is poor, with a median time from IPF diagnosis to death of only 2–3 years [5]. According to incomplete statistics, the incidence of IPF is slightly higher in North America and Europe compared to South America and East Asia [6].

It has been suggested that factors such as repeated epithelial cell injury, myofibroblast recruitment and activation, and fibroblast differentiation may be involved in the pathogenesis of IPF [7,8]. In recent years, many genes have been implicated in the pathogenesis of IPF. However, the underlying molecular pathways of IPF remain elusive. Thus, the molecular mechanisms underlying the development of IPF need to be urgently elucidated. This is likely to provide better targets for finding therapeutic agents and facilitating therapeutic modalities.

Large-scale data analysis using gene sequencing, bioinformatics, and microarray technology has been widely used to identify disease signatures and potential biomarkers [9]. Uncertain genome sequences of an individual can be identified with gene sequencing technology, while bioinformatics can process large amounts of genome sequence information [10,11]. The identification of differentially expressed genes (DEGs) in healthy controls and IPF patients, based on the analysis of the Gene Expression Omnibus (GEO) database [12], provides an avenue for a more effective understanding of the underlying molecular mechanisms of IPF pathogenesis.

The GSE110147 and GSE53845 datasets were used for this study. DEGs between IPF lung samples and healthy controls were identified using bioinformatics techniques. The goal of this study was to elucidate the genetic pathogenesis of IPF and to obtain new clues for the discovery of therapeutic agents and the advancement of therapeutic approaches.

## 2. Results

### 2.1. The Identification of DEGs

From the GEO database, the datasets GSE110147 and GSE53845 were picked. Table 1 and Figure 1 show the features of normal and IPF lung samples from the two GEO datasets. The normal and IPF samples were nicely differentiated. Significant DEGs were observed between the two groups (Figure 1A,D). After standardization of the microarray results, analysis was performed using the GEO2R. The sample distributions of the two datasets were consistent, with medians on the same line in Figure 1B,E. All expressed genes are shown as volcano plots in Figure 1C,F. As displayed in the Venn diagram (Figure 1G) between IPF and normal lung tissue, the overlapping of the two datasets included 240 genes.

### 2.2. Gene Ontology (GO) and Kyoto Encyclopedia of Genes and Genomes (KEGG) Enrichment Analysis

According to the GO analysis results, changes in biological processes (BP) of DEGs were strongly enriched in positive regulation of smooth muscle cell migration, cell adhesion, extracellular matrix organization, angiogenesis, cellular response to fibroblast growth factor stimulus, and positive regulation of cell migration (Figure 2A). As shown in Figure 2A, changes in molecular function (MF) of DEGs were focused on heparin binding, protein dimerization activity, extracellular matrix structural constituent conferring tensile strength, extracellular matrix structural constituent, calcium ion binding, integrin binding, serine-type endopeptidase activity, and identical protein binding. As presented in Figure 2A, changes in the cell component (CC) of DEGs were mostly enriched in cell surface, extracellular space, extracellular region, integral component of membrane, extracellular matrix, plasma membrane, and basement membrane. Based on the KEGG pathway analysis results, the DEGs were mainly enriched in ether lipid metabolism, focal adhesion, mineral absorption, ECM–receptor interaction, cell adhesion molecules, complement and coagulation cascades, ovarian steroidogenesis, and protein digestion and absorption (Figure 2B).

### 2.3. Protein–Protein Interaction (PPI) Network

As shown in Figure 3A, the STRING database was used to construct the PPI network for the DEGs. Cytoscape was used for the main modules. Then, via a computational method called Molecular Complex Detection (Mcode), we detected and analyzed four important modules in the PPI network (Figure 3B–E). Specifically, the first module contained 14 potential hub genes: *VCAM1*, *CCL5*, *LEPREL1*, *COL17A1*, *COL1A1*, *FAP*, *ASPN*, *HIF1A*, *CDH2*, *SPP1*, *MMP7*, *MMP13*, *MMP1*, and *POSTN* (Figure 3B).

### 2.4. Hub Gene Selection

The Cytoscape plug-in cytoHubba, which facilitates the identification of hub genes, was used to evaluate the PPI network. Based on the Maximum Clique Centrality (MCC) algorithm, the top 10 genes, *COL1A1*, *MMP1*, *SPP1*, *VCAM1*, *IGF1*, *POSTN*, *MMP7*, *CDH2*, *COL3A1*, and *MMP13*, were selected as potential hub genes (Figure 4A). By overlapping the hub genes derived from cytoHubba with the hub genes derived from Mcode, we obtained eight hub genes: *VCAM1*, *COL1A1*, *CDH2*, *SPP1*, *MMP7*, *MMP13*, *MMP1*, and *POSTN* (Figure 4B). The characteristics of the hub genes are shown in Table 2. Using GeneMANIA, the co-expression networks and possible functionalities of the hub genes were investigated (Figure 4C). They discovered sophisticated PPI networks with a co-expression of 45.32%, genetic interactions of 0.05%, co-localization of 2.53%, protein domains of 15.88%, prediction of 20.12%, and physical interactions of 16.11%. Functional assessment revealed that they were mainly involved in a variety of collagen metabolism pathways as well as extracellular matrix organization, including serine-type peptidase activity, metallopeptidase activity, extracellular matrix organization, collagen metabolism process, response to UV-A, serine hydrolase activity, and cellular response to UV, revealing their essential role in contributing to IPF pathogenesis. The eight hub genes from both datasets were displayed using the heatmaps, as shown in Figure 5A,B.

### 2.5. Increased Expression of Hub Gene in IPF Lung Tissues

Figure 6A–H show the expression of SPP1, VCAM1, CDH2 and POSTN by GSE110147 and GSE53845, respectively. Mice in the WT bleomycin (BLM) group showed disorganized alveolar structure, thickened septa, increased collagen deposition, and aggregated monocyte infiltration (Figure 7A). The mRNA expression levels of SPP1, VCAM1, CDH2, and POSTN were higher in the WT BLM group compared to the WT saline group, consistent with our bioinformatics predictions (Figure 7B). Immunohistochemistry (IHC) results showed that SPP1, VCAM1, CDH2, and POSTN were highly expressed in the WT BLM group, in agreement with our bioinformatics predictions (Figure 7C).

### 2.6. Construction of a microRNA (miRNA)-Gene Network

To estimate the target miRNAs of the hub genes, the NetworkAnalyst databases were used. The miRNA-gene interaction network containing three seeds, 116 nodes and 116 edges was generated using Cytoscape software (version 3.9.1). Two common miRNA targets (miR-181b-5p and miR-4262) were found to interact with *SPP1* and *VCAM1*, as shown in Figure 8. *CDH2* and *VCAM1* were able to interact with one common target miRNA: miR-155-5p (Figure 8). Nevertheless, this remains to be seen and needs further validation.

## 3. Discussion

IPF generally has a poor prognosis and limited treatment options [13]. In most countries, there has been an increase in the incidence of IPF over time [14]. Bioinformatics technology is widely used in medicine to find disease-causing genes and biomarkers [10]. The pathogenesis of IPF has been a mystery for many years. Therefore, this work aims to elucidate the critical DEGs and signaling pathways in IPF based on bioinformatics approaches.

In this study, the GSE110147 and GSE53845 datasets were selected to screen 240 common DEGs. Eight genes *(VCAM1*, *COL1A1*, *CDH2*, *SPP1*, *MMP7*, *MMP13*, *MMP1*, *POSTN*) were selected as hub genes. The DEGs (*COL1A1*, *MMP7*, *MMP13*, *MMP1*) were mainly mapped in fibrotic diseases. Therefore, the present study focused on the remaining four hub genes (*VCAM1*, *CDH2*, *SPP1*, *POSTN*). Additionally, the IHC and RT-qPCR results of lung tissue from a mouse model further confirmed these results. This suggests that DEGs may play a role in causing IPF.

In our report, enriched GO terms and KEGG pathways revealed a large number of differences between IPF and normal lung tissues. Then, GO enrichment analysis revealed that a large variety of biologically relevant processes associated with positive regulation of smooth muscle cell migration, cell adhesion, cellular response to fibroblast growth factor stimulus, angiogenesis, positive regulation of cell migration, and extracellular matrix organization were significantly enriched. Previous studies have clarified that changes in cell adhesion [15], positive regulation of smooth muscle cell migration, cellular response to fibroblast growth factor stimulus, positive regulation of cell migration, extracellular matrix organization [16], and angiogenesis [17] may affect the development of fibrotic disease [18,19]. Furthermore, KEGG enrichment analysis showed some correlation with cell adhesion molecules, protein digestion and absorption, focal adhesion, and ECM–receptor interaction. In IPF, cell adhesion molecules are involved in leukocyte recruitment and activation, mediate cell–cell interactions between cells of the same type, and mediate cell interactions with extracellular matrix molecules [20]. A typical signaling pathway in fibrotic diseases is the ECM–receptor interaction [21]. By stimulating cell spreading and migration, focal adhesions drive the progression of pulmonary fibrosis [22]. Thus, our conclusions from the KEGG enrichment analysis are in agreement with earlier results [23].

Subsequently, we structured the PPI network of DEGs. With Mcode’s computational approach, we discovered four vital modules from this network. Specifically, the first modules contained 14 potential hub genes. The top 10 genes were selected using the MCC algorithm. Taking the intersection of the genes by the two methods, the following eight hub genes were obtained: *VCAM1*, *COL1A1*, *CDH2*, *SPP1*, *MMP7*, *MMP13*, *MMP1*, and *POSTN*. The DEGs (*COL1A1*, *MMP7*, *MMP13*, *MMP1*) were mainly mapped in fibrotic diseases. Proteases that degrade all components of the extracellular matrix, matrix metalloproteinases (MMPs) [24]. Plasma MMP7 levels have been shown to be a biomarker for IPF [25]. The fibrotic response of the lung to injury is differentially influenced by MMP13 [24]. However, MMP1 protects against IPF [24]. In particular, COL1A1 is a myofibroblast marker [26]. Therefore, the present study focused on the remaining four hub genes (*VCAM1*, *CDH2*, *SPP1*, *POSTN*).

VCAM1 mediates leukocyte adhesion to the vascular endothelium [27]. VCAM1 protein and mRNA levels were found to be higher in IPF lungs than in control lungs [28]. Further studies revealed that depletion of VCAM1 inhibited fibroblast proliferation. Thus, VCAM1 is implicated in promoting the progression of fibrotic disease in IPF patients. *CDH2* encodes the N-cadherin protein, a member of the adhesin family of proteins [29]. During epithelial–mesenchymal transition, epithelial cells become more aggressive, with upregulation of N-cadherin, which is responsible for fibrosis [30]. *SPP1* encodes the osteopontin (OPN) protein, currently recognized as a key cytokine that contributes to immune cell recruitment [31]. OPN regulates tissue repair and remodeling [32], and helps fibroblasts adhere, migrate, and proliferate [33]. *POSTN* encodes the periostin protein, which maintains extracellular matrix homeostasis [34]. In this paper, the results were confirmed using IHC and RT-qPCR analysis of lung tissue from a mouse model.

MiRNAs degrade or inhibit translation by binding to target mRNAs and negatively regulating gene expression [35]. We performed a miRNA-gene network construction in this project. Two common miRNA targets were found to interact with *SPP1* and *VCAM1*. *CDH2* and *VCAM1* could interact with one common target miRNA. Previous studies on miR-181b-5p and miR-4262 have mainly focused on tumors. However, miR-181b-5p and miR-4262 have not been reported in IPF. Significantly expressed hsa-miR-181b-5p and miRNA pathogenesis in childhood acute lymphoblastic leukemia were confirmed by studies [36]. Studies have shown that miR-4262 is a potential tumor promoter that promotes the proliferation and invasive ability of cancer cells in human cancers [37]. MiR-155-5p reduces renal interstitial fibrosis through promoting autophagy [38]. Another study reported that miR-155-5p exacerbates alveolitis by targeting FOXO3a and promotes pulmonary fibrosis [39].

In conclusion, this paper used bioinformatics analysis to obtain some hub genes. The analysis revealed that miRNAs are involved in the pathogenesis of IPF by interacting with hub genes. However, this study has some shortcomings. First, the almost inevitable heterogeneity between different data sets, data platforms, and statistical analyses may affect the reliability of this study. Second, it did not fully review a large enough number of IPF-related datasets. Only two GSE datasets were analyzed in our report. Finally, detailed studies on the regulation of these hub genes and miRNAs in IPF are lacking. Despite these limitations, new understanding of the development of IPF may be gained from this study.

## 4. Materials and Methods

### 4.1. Data Download

From the NCBI GEO (http://www.ncbi.nlm.nih.gov/geo, accessed on 20 February 2023) [40] datasets GSE110147 [11] (excluding non-specific interstitial pneumonia samples) and GSE53845 [41], which contain clinical information on IPF and normal lung tissue.

### 4.2. DEGs Screening and Data Processing

To identify DEGs in IPF compared to normal tissue, GEO2R (http://www.ncbi.nlm.nih.gov/geo/geo2r, accessed on 20 February 2023) was applied (|logFC| ≥ 1 and adj. *p* ≤ 0.01). The boxplot was plotted by the Xiantao website (https://www.xiantao.love/, accessed on 20 February 2023). The Jvenn online tools were used to create the Venn diagrams (http://www.bioinformatics.com.cn/static/others/jvenn/example.html, accessed on 21 February 2023) [42].

### 4.3. KEGG and GO Enrichment Analysis

DAVID (http://david.ncifcrf.gov, accessed on 22 February 2023) (DAVID version 2021) [43,44] is a database that collects biological data and analysis tools. The DAVID database was used for biological analysis (KEGG [45] and GO [46] enrichment analysis). *p* < 0.05 was considered statistically significant.

### 4.4. PPI Network

The PPI network was created using the STRING database (http://string-db.org, accessed on 23 February 2023) [47] (version 11.5). An interaction was considered statistically significant if the combined score was >0.4. A platform for the visualization of molecular interaction networks is Cytoscape (version 3.9.1) [48].

### 4.5. Hub Genes

To find areas of dense connections, Cytoscape’s Mcode plugin [49] (version 2.0.2) is used. The criteria for selection were as follows: Mcode scores > 3, degree cut-off = 2, node score cut-off = 0.2, Max depth = 100 and k-score = 2. Each node gene is scored by Cytoscape’s cytoHubba plugin using MCC. To screen for pivot genes, the top 10 hub genes of each algorithm’s MCC score were used. GeneMANIA (http://genemania.org, accessed on 24 February 2023) generated gene function predictions and gene maps with comparable effects [50]. To identify PPI networks of eigengenes, we used GeneMANIA. Violin plot was plotted by http://www.bioinformatics.com.cn/plot_basic_ggviolin_plot_113 (accessed on 25 February 2023). Heatmap was plotted by http://www.bioinformatics.com.cn/plot_basic_cluster_heatmap_plot_024 (accessed on 25 February 2023).

### 4.6. Animals and BLM-Induced Mouse Model

C57BL/6 males (8–10 weeks old) were obtained from the Laboratory Animal Center of Xi’an Jiaotong University. They were maintained in a standard animal facility with a room temperature of 25 °C, a 12-h light/dark cycle, and food and water were freely available. The mice in the experiments strictly followed the Guidelines for the Care and Use of Laboratory Animals and were approved by the Biomedical Ethics Committee of Health Science Center of Xi’an Jiaotong University (No. XJTUAE2023-1301, 27 February 2023).

A total of 30 male mice were randomly assigned to two groups, namely: (1) WT saline group, (2) WT BLM group. BLM (Haizhenghuirui, Fuyang, China) was dissolved in saline. Mice in the experimental group were administered BLM (intraperitoneal injection, 10 mg/kg), and the control group was administered saline. The injections were administered for 10 consecutive days. After their weight was measured, samples were taken on day 29 of the last dose.

### 4.7. Collection of Lung Tissue

Lung tissue was rinsed with ice-cold PBS, the right lung tissue was frozen in a −80 °C refrigerator, and the left lung lobe was fixed with 4% paraformaldehyde for pathological staining.

### 4.8. Pathological Staining

Lung tissues were embedded in paraffin and sectioned at 4 μm. The sections were stained with H&E or Masson’s trichrome kit (Nanjing Jiancheng Co., Ltd., Nanjing, China). For IHC analysis of SPP1, VCAM1, CDH2, and POSTN expression in the WT saline group and WT BLM group, sections were incubated overnight at 4 °C with the appropriate primary antibodies as follows: immunohistochemical analysis of anti-SPP1 (1:80; Zen-Bio Science, Chengdu, China), anti-VCAM1 (1:200; Santa Cruz, CA, USA), anti-CDH2 (1:2000; ProteinTech Group, Inc., Rosemont, IL, USA), and anti-POSTN (1:200; Santa Cruz, CA, USA). The detection system used was a DAB kit from CWBIO Co., Ltd. (Beijing, China). Slides were counterstained using hematoxylin. At least 4 biological replicates were performed for each group. There were 3 technical replicates were performed for each lung tissue sample.

### 4.9. RNA Extraction and Quantitative Polymerase Chain Reaction (qPCR)

Using TRIzol reagent (cat. no. RK30129, ABclonal, Woburn, MA, USA), RNA was extracted from lung tissues. cDNA was obtained using a reverse transcription kit (cat. no. RK20428, ABclonal, Woburn, MA, USA). The primer sequences are as follows: M-VCAM1 (forward, 5′-TACTGTTTGCAGTCTCTCAAGC-3′; reverse, 5′-CAAGTGAGGGCCATGGAGTC-3′), M-CDH2 (forward, 5′-GGCCTTGCTTCAGGCGT-3′; reverse, 5′-CATTGAGAAGGGGCTGTCCT-3′), M-SPP1 (forward, 5′-CCTGGCTGAATTCTGAGGGAC-3′; reverse, 5′-ATCAGTCACTTTCACCGGGAG-3′), M-POSTN (forward, 5′-GAAGTGATCCACGGAGAGCC-3′; reverse, 5′-CCTCCTGTGGAAATCCTGGT-3′), and M-GAPDH (forward, 5′- GCACCGTCAAGGCTGAGAAC-3′; reverse 5′- TGGTGAAGACGCCAGTGGA-3′). At least 6 biological replicates were performed for each group. There were 3 technical replicates were performed for each lung tissue sample.

### 4.10. MiRNAs Associated with Hub Genes

To construct the miRNA-gene interactions of the hub genes, the NetworkAnalyst tool (version 3.0, https://www.networkanalyst.ca/, accessed on 26 February 2023) was used.

### 4.11. Statistical Analysis

Image J and GraphPad Prism software for Windows (v8.0, San Diego, CA, USA) were used for all statistical analyses. Differences between the two groups were evaluated using Student’s *t*-test. *p* < 0.05 was defined as statistically significant. All experiments were performed more than three times.

## 5. Conclusions

Eight hub genes have been implicated in the pathogenesis of IPF. Four of the hub genes (*VCAM1*, *CDH2*, *SPP1*, and *POSTN*) were validated in animal experiments and were significantly upregulated. Finally, our novel miRNA-gene network provided new insights into the deeper mechanisms of IPF. MiR-181b-5p, miR-4262, and miR-155-5p may be involved in the pathophysiological processes of IPF by interacting with hub genes.

## Figures and Tables

**Figure 1 ijms-24-13305-f001:**
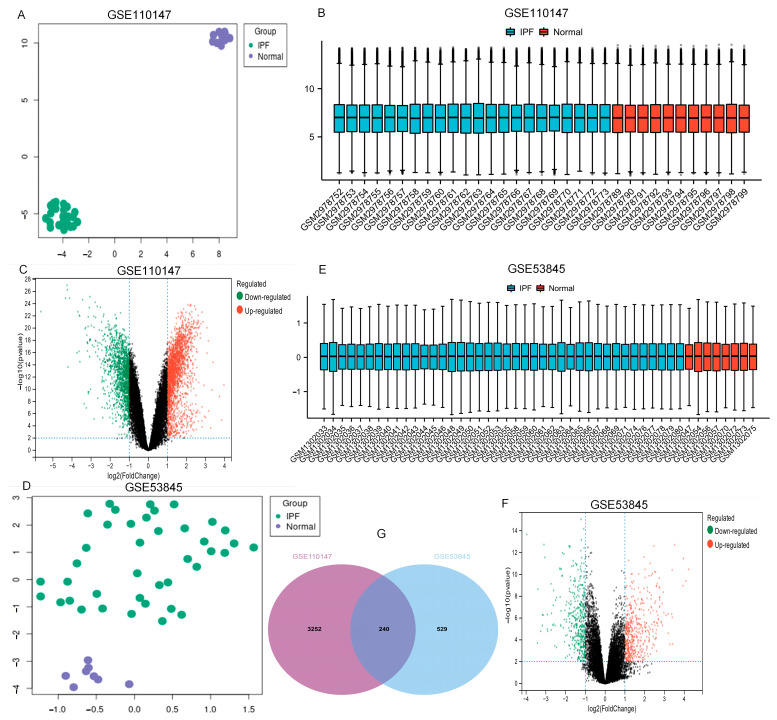
UMAP, boxplot, and volcano plots of the two datasets (|logFC| ≥ 1 and adj. *p* ≤ 0.01). (**A**–**C**) GSE110147, (**D**–**F**) GSE53845, and (**G**) Venn diagrams. Abbreviations: IPF, idiopathic pulmonary fibrosis.

**Figure 2 ijms-24-13305-f002:**
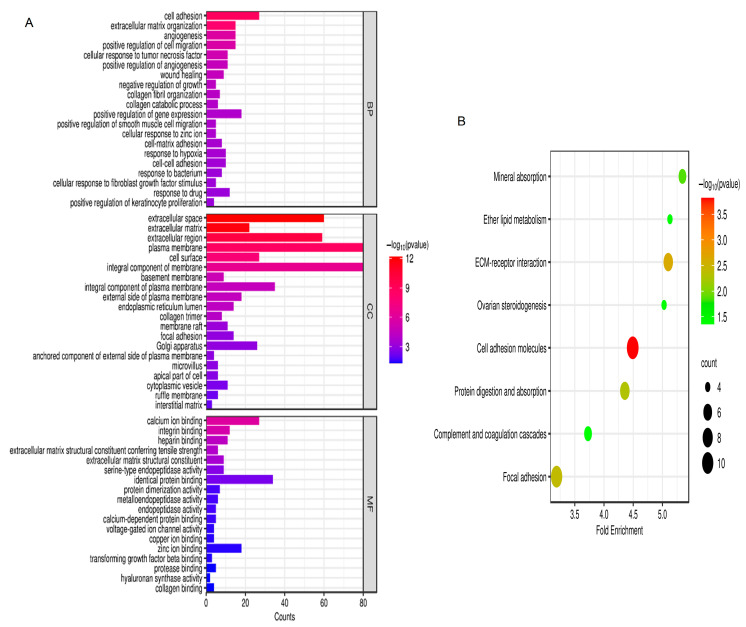
GO and KEGG analysis. (**A**) GO analysis; (**B**) KEGG analysis. *p* < 0.05. Abbreviations: BP, biological process; CC, cell component; MF, molecular function; GO, Gene Ontology; KEGG, Kyoto Encyclopedia of Genes and Genomes.

**Figure 3 ijms-24-13305-f003:**
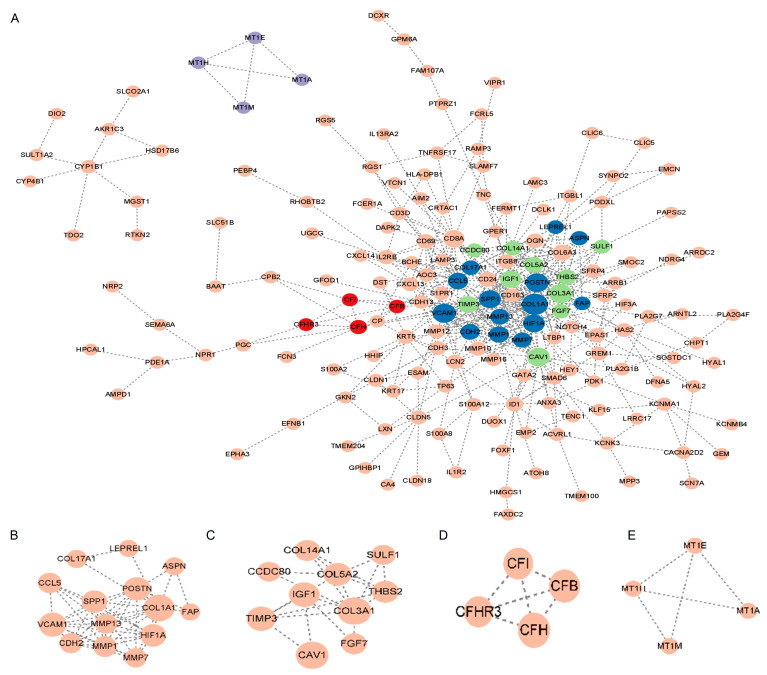
PPI network. (**A**) PPI network of major DEGs. The node in dark blue represents module B, the node in green represents module C, the node in red represents module D, and the node in purple represents module E. (**B**–**E**) Important PPI network modules. Abbreviations: PPI, Protein–protein interaction; DEGs, differentially expressed genes.

**Figure 4 ijms-24-13305-f004:**
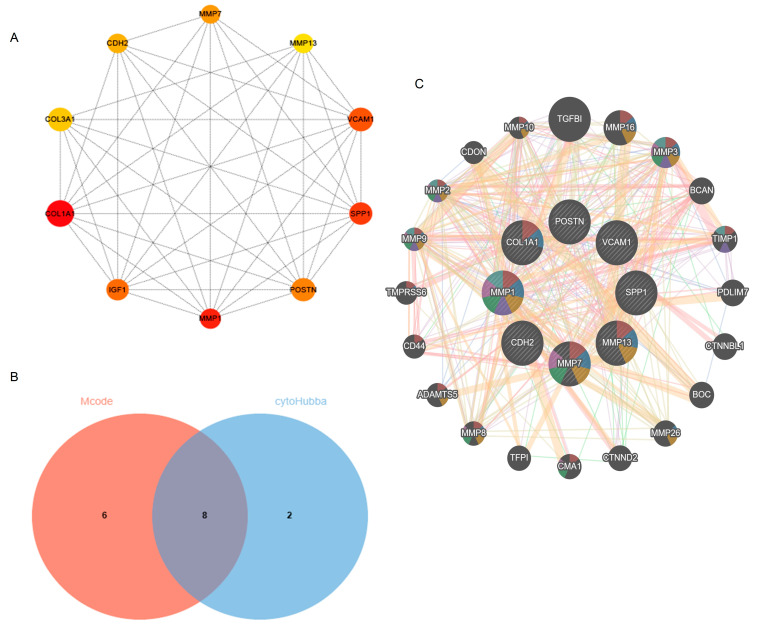
Identification of hub genes using cytoHubba and Mcode. (**A**) cytoHubba’s top 10 hub gene PPI network. (**B**) Venn diagram. (**C**) Hub genes and their co-expression genes. Abbreviations: Mcode, Molecular Complex Detection; PPI, Protein–protein interaction.

**Figure 5 ijms-24-13305-f005:**
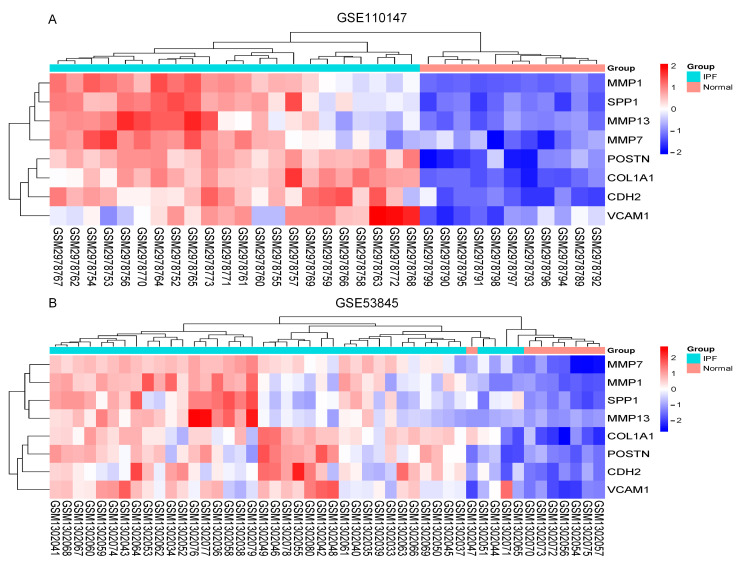
The heatmaps. (**A**) Heatmap in GSE110147; (**B**) Heatmap in GSE53845. Abbreviations: IPF, idiopathic pulmonary fibrosis; MMP13, Matrix Metallopeptidase 13; SPP1, Secreted Phosphoprotein 1; MMP7, Matrix Metallopeptidase 7; VCAM1, Vascular cell adhesion molecule 1; CDH2, Cadherin 2; POSTN, Periostin; MMP1, Matrix Metallopeptidase 1; COL1A1, Collagen Type I Alpha 1 Chain.

**Figure 6 ijms-24-13305-f006:**
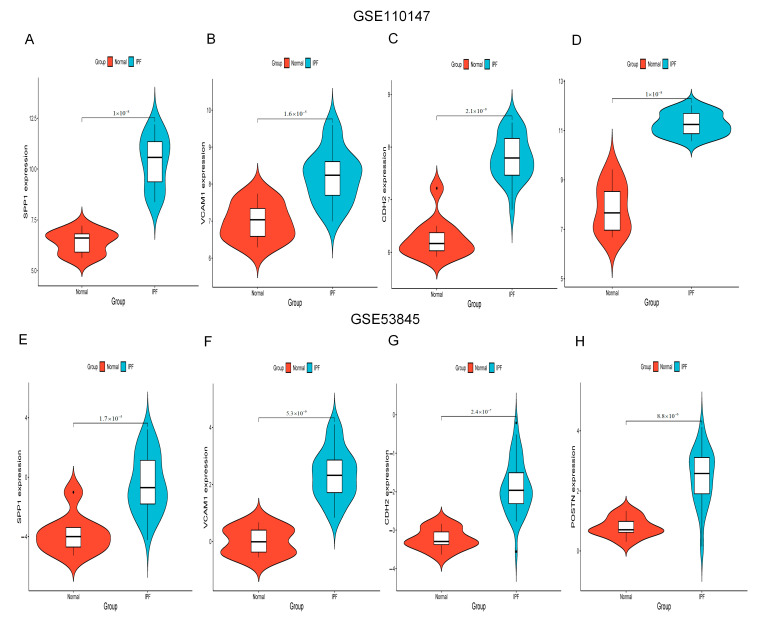
SPP1, VCAM1, CDH2, and POSTN expression. (**A–D**) SPP1, VCAM1, CDH2, and POSTN expression by GSE110147. (**E–H**) SPP1, VCAM1, CDH2, and POSTN expression by GSE53845. Abbreviations: IPF, idiopathic pulmonary fibrosis; VCAM1, Vascular cell adhesion molecule 1; SPP1, Secreted Phosphoprotein 1; CDH2, Cadherin 2; POSTN, Periostin.

**Figure 7 ijms-24-13305-f007:**
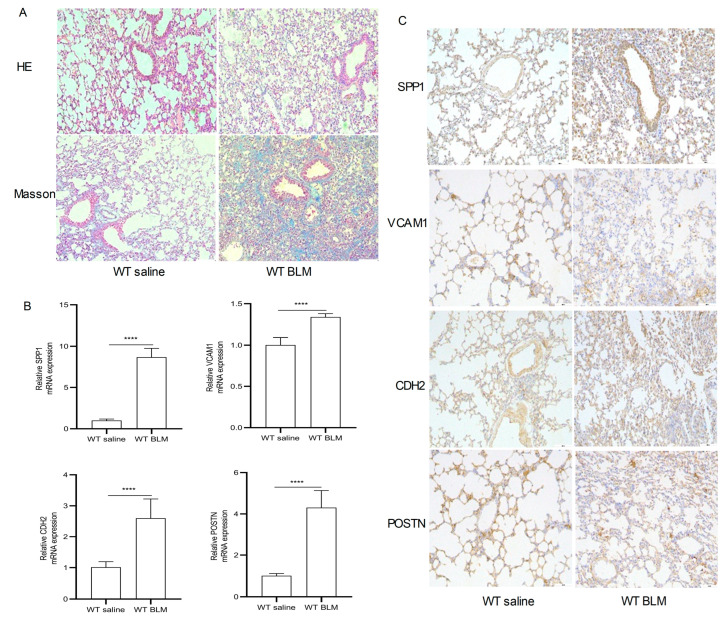
Pathological staining and RT-qPCR. (**A**) Hematoxylin–eosin (HE) and Masson Trichrome staining in the WT saline group and WT BLM group (original magnification, ×100). (**B**) RT-qPCR analysis levels of SPP1, VCAM1, CDH2, and POSTN mRNA expression in the WT saline group and WT BLM group (data represent the mean ± SD, n = 6). **** *p* < 0.0001. (**C**) Immunohistochemical analysis of SPP1, VCAM1, CDH2, and POSTN expression in the WT saline group and WT BLM group (original magnification, ×100, n = 4). Abbreviations: BLM, bleomycin; SPP1, Secreted Phosphoprotein 1; VCAM1, Vascular cell adhesion molecule 1; CDH2, Cadherin 2; POSTN, Periostin.

**Figure 8 ijms-24-13305-f008:**
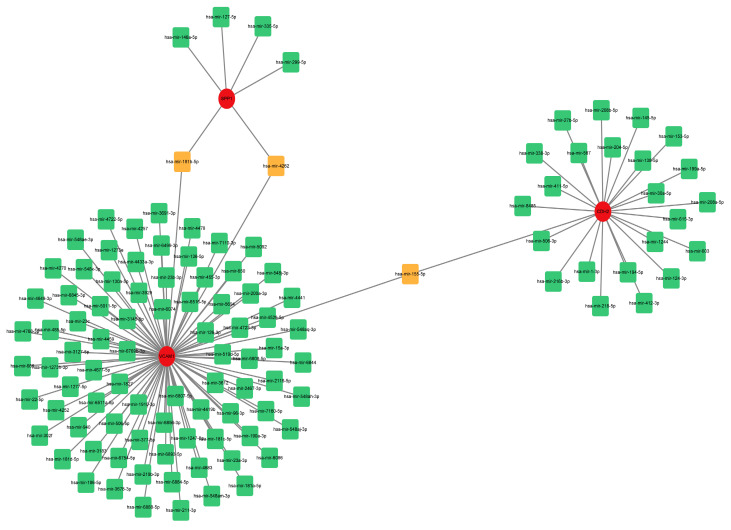
Comprehensive network of miRNA-gene interactions for the three hub genes. Green and orange squares represent miRNA, and red circles represent three hub genes. Abbreviations: miRNA, microRNA.

**Table 1 ijms-24-13305-t001:** Features of the GSE datasets.

Data Set	Platforms	Sample	Normal	IPF	General Information
GSE110147	GPL6244	33	11	22	disease state, tissue
GSE53845	GPL6480	48	8	40	disease state, tissue, diagnosis, source, gender, sample type

**Table 2 ijms-24-13305-t002:** The biological function of biomarkers in detail from GeneCards database.

NO.	Gene Symbol	Full Name	Function
1	MMP13	Matrix Metallopeptidase 13	Plays a role in the degradation of extracellular matrix proteins including fibrillar collagen, fibronectin, TNC and ACAN.
2	SPP1	Secreted Phosphoprotein 1	Major non-collagenous bone protein that binds tightly to hydroxyapatite. Appears to form an integral part of the mineralized matrix. Probably important to cell–matrix interaction.
3	MMP7	Matrix Metallopeptidase 7	Degrades casein, gelatins of types I, III, IV, and V, and fibronectin. Activates procollagenase.
4	VCAM1	Vascular Cell Adhesion Molecule 1	Cell adhesion glycoprotein predominantly expressed on the surface of endothelial cells that plays an important role in immune surveillance and inflammation. Acts as a major regulator of leukocyte adhesion to the endothelium through interaction with different types of integrins.
5	CDH2	Cadherin 2	Calcium-dependent cell adhesion protein; preferentially mediates homotypic cell–cell adhesion by dimerization with a CDH2 chain from another cell. Cadherins may thus contribute to the sorting of heterogeneous cell types.
6	COL1A1	Collagen Type I Alpha 1 Chain	Type I collagen is a member of group I collagen (fibrillar forming collagen).
7	POSTN	Periostin	Induces cell attachment and spreading and plays a role in cell adhesion. Enhances incorporation of BMP1 in the fibronectin matrix of connective tissues, and subsequent proteolytic activation of lysyl oxidase LOX (By similarity).
8	MMP1	Matrix Metallopeptidase 1	Cleaves collagens of types I, II, and III at one site in the helical domain. Also cleaves collagens of types VII and X.

## Data Availability

Data will be made available on request. The data that support the findings of this study are available from the corresponding author, Dr. Liu, upon reasonable request.

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
