# Peer review of "Underlying Molecular Mechanism and Construction of a miRNA-Gene Network in Idiopathic Pulmonary Fibrosis by Bioinformatics"

_ijms, 2023, doi:10.3390/ijms241713305_

Round 1
Reviewer 1 Report
Dear authors
I have studied with great interest the manuscript ‘’The underlying molecular mechanism in idiopathic pulmonary fibrosis by bioinformatics and validation’’
The research question and the topics studied are of great interest. The authors have performed a bioinformatics study to identify differentially expressed genes (DEGs) in idiopathic pulmonary fibrosis and subsequently performed an in vivo validation with a mouse model of this disease. Experimental data are adequate although relevant information related to the study design (technical replicates, biological replicates) is missing. The grammar needs to be checked and it should be check -spelled.
I have some comments:
-Review abbreviations. These are described the first time they appear in the manuscript and thereafter refer to them only with the abbreviation (e.g., differentially expressed genes is defined in line 57 and 61).
-The title does not adequately indicate the content of the article. It should be more specific
-The abstract is too long and starts talking about material and methods used. It should start with a brief introduction and I recommend summarize the abstract to 200-250 words.
-The figures are of poor quality. Especially figures 3, 4 and 8 do not have good resolution, they do not read clearly. In addition, there are too many nodes, which are not necessary or relevant, I recommend reducing the PPi networks and showing only the important nodes, with less background and higher resolution, larger font size and greater clarity and cleanliness. I also recommend defining the abbreviations used in the figures in the figure captions to facilitate the reading of the manuscript.
-Section 4.6 should indicate the number of animals included in the experiments, both in the control group and in the experimental group. In addition, the technical replicates and biological replicates should be indicated.
Please revise for language and grammar.Change 'analyze' to analyze throughout the manuscript. Check the sentence structure.
Reviewer 2 Report
ijms-2557425
The authors used bioinformatic methods to study the differentially expressed genes (DEG) of human idiopathic pulmonary fibrosis (IPF) and to elucidate the pathogenesis of IPF at the genetic level- microarrays GSE110147 and GSE53845 were downloaded from the Gene Expression Omnibus (GEO) database and analyzed. to obtain DEGs between IPF and normal lung tissue with the GEO2R online tool. Additional studies performed on lung tissue from a mouse model of bleomycin-induced interstitial lung fibrosis were validated by immunohistochemistry (IHC) and RT-qPCR. The authors demonstrate that four of the hub genes (VCAM1, CDH2, SPP1 and POSTN) have been validated in animal experiments and are significantly upregulated. Some potential target miRNAs, including miR-181b-5p, miR-4262, and miR-155-5p, may be involved in the pathophysiological processes of IPF by interacting with hub genes.
The manuscript is at a high scientific level and will certainly be applicable in practice; Minor remarks: 1.The conclusion should be rewritten and attention should be paid to the merits of the study; 2. Where possible references used will be updated.
Minor editing of English language required
Round 2
Reviewer 1 Report
Dear authors,
Thank you very much for following my suggestions, the manuscript has improved its quality and scientific soundness and will undoubtedly be of great interest to deepen the knowledge of the molecular mechanisms of idiopathic pulmonary fibrosis and to be able to develop diagnostic or therapeutic strategies directed to it.
I just have one more question
Is it necessary for so many nodes to appear in Figure 3A and Figure 8?
The fact that so many nodes appear makes it more difficult to highlight which nodes are the most relevant.
If it is necessary that they all appear, perhaps a color code could be made to locate the relevant nodes.
Thank you for your work,
Look forward to hearing from you
Regards
